# The Antihypertensive Effects and Safety of LCZ696 in Patients with Hypertension: A Systemic Review and Meta-Analysis of Randomized Controlled Trials

**DOI:** 10.3390/jcm10132824

**Published:** 2021-06-26

**Authors:** Su-Kiat Chua, Wei-Ting Lai, Lung-Ching Chen, Huei-Fong Hung

**Affiliations:** 1School of Medicine, College of Medicine, Fu Jen Catholic University, New Taipei 24205, Taiwan; M006507@ms.skh.org.tw; 2Division of Cardiology, Department of Internal Medicine, Shin Kong Wu Ho-Su Memorial Hospital, Taipei 111045, Taiwan; M011645@ms.skh.org.tw (W.-T.L.); M010281@ms.skh.org.tw (L.-C.C.); 3Department of Internal Medicine, Shin Kong Wu Ho-Su Memorial Hospital, Taipei 111045, Taiwan

**Keywords:** LCZ696, ambulatory systolic blood pressure, hypertension

## Abstract

Background: The management of hypertension remains suboptimal throughout the world. Methods: We performed a random-effects model meta-analysis of randomized controlled trials to determine the effectiveness and safety of sacubitril/valsartan (LCZ696) for the treatment of high arterial pressure. Relevant published articles from PubMed, Cochrane base, and Medline were examined, and the last search date was December 2020. Only published randomized controlled trials and double-blind studies were selected for further analysis. The mean reductions in systolic blood pressure (msSBP) and diastolic blood pressure (msDBP) in the sitting position, as well as the mean reductions in ambulatory systolic blood pressure (maSBP) and ambulatory diastolic blood pressure (maDBP), were assumed as efficacy endpoints. Adverse events (AEs) were considered as safety outcomes. Results: Ten studies with a total of 5931patients were included for analysis. Compared with placebo, LCZ696 had a significant reduction in msSBP (weight mean difference (WMD) = −6.52 mmHg, 95% confidence interval (CI): −8.57 to −4.47; *p* < 0.001), msDBP (WMD = −3.32 mmHg, 95% CI: −4.57 to −2.07; *p* < 0.001), maSBP (WMD = −7.08 mmHg, 95% CI: −10.48 to −3.68; *p* < 0.001), maDBP (WMD = −3.57 mmHg, 95% CI: −5.71 to −1.44, *p* < 0.001). In subgroup analysis, only 200 mg and 400 mg LCZ696 showed a significant BP reduction. There was no difference in the AE rate between the LCZ696 and placebo groups (WMD = 1.02, 95% CI: 0.83 to 1.27, *p* = 0.54). Egger’s test revealed a potential publication bias for msSBP (*p* = 0.025), but no publication bias for other outcomes. Conclusion: LCZ696 may reduce blood pressure more efficaciously than traditional therapy in hypertensive patients without increasing adverse effects.

## 1. Introduction

Hypertension is a well-known and modifiable risk factor for cardiovascular disease worldwide. A study showed that the number of patients will increase to 1.5 billion by 2025 [1]. However, the management of hypertension remains suboptimal throughout the world [2,3]. Sacubitril/valsartan (LCZ696), a first-in-class angiotensin receptor neprilysin inhibitor, showed superior benefits over enalapril in the PARADIGM-HF trial [4] and was approved for the treatment of heart failure with reduced ejection fraction. Neprilysin inhibition results in blood pressure reduction via natriuresis, vasodilatation, and inhibition of the renin-angiotensin-aldosterone system and sympathetic activity [5]. The dual inhibition of neprilysin and the angiotensin receptor has shown complementary effects in blood pressure reduction. Omapatrilat resulted in superior effects on systolic blood pressure and pulse pressure reduction than enalapril [6], but was not approved due to the unacceptable rate of angioedema.

In recent years, several randomized controlled trials (RCTs) have attempted to compare the antihypertensive efficacy and safety of LCZ696 in hypertensive patients [7,8,9,10,11,12,13,14,15,16]. Most of these prospective and double-blind clinical trials showed that LCZ696 achieved the target blood pressure better without significant adverse effects. In addition, several meta-analyses have concluded that LCZ696 may reduce arterial pressure more efficaciously than placebo, particularly the angiotensin receptor blocker (ARB), without increasing overall adverse events [17,18,19]. However, some meta-analyses included non-RCTs, retrospective articles, or unpublished studies [17,18]. We performed a meta-analysis including high quality RCTs to precisely determine the effectiveness and safety of LCZ696 for the treatment of high arterial pressure.

## 2. Materials and Methods

### 2.1. Study Selection, Search Strategy, and Outcome Measures

This systematic review and meta-analysis adhered to the Preferred Reporting Items for Systematic Reviews and Meta-Analysis (PRISMA) guidelines [20]. The inclusion criteria for this network meta-analysis were as follows: (1) all relevant Phase 3 RCTs comparing LCZ696 and placebo in patients with hypertension; (2) reported mean systolic blood pressure (msSBP) and mean diastolic blood pressure (msDBP) in the sitting position; (3) reported mean ambulatory systolic blood pressure (maSBP) and mean ambulatory diastolic blood pressure (maDBP); (4) reported trial-defined adverse events; and (5) a follow-up period of at least 6 months. The primary exclusion criteria were observational studies, registry data, editorials, case series, crossover trials, duplicate studies, and non-original data.

We searched PubMed, Embase, and the Cochrane database throughout December of 2020 using the following key words in various combinations: “hypertension”, “high blood pressure”, “angiotensin receptor blockers”, “angiotensin receptor antagonists”, “angiotensin II receptor antagonists”, “angiotensin II receptor blockers”, “LCZ696”, “ARNI”, and “angiotensin receptor neprilysin inhibitor” and “randomized clinical trial”. Two investigators (SKC and LCC) independently reviewed the titles or abstracts of the studies to determine whether they met the inclusion and exclusion criteria. Disagreements were resolved via consensus and by a third investigator (LCC).

### 2.2. Data Synthesis and Statistical Analyses

The following details were gained from all the included studies and were assumed as efficacy endpoints: patient characteristics, sex ratio, enrolled numbers, LCZ696 drug dose, mean reductions in systolic blood pressure (msSBP) and diastolic blood pressure (msDBP) in the sitting position, as well as mean reductions in ambulatory systolic blood pressure (maSBP) and ambulatory diastolic blood pressure (maDBP). The main outcomes were the difference in the means of the blood pressure between LCZ696 and traditional therapy (or placebo). Adverse events (AEs) were considered as safety outcomes. The summary adjusted odds ratio (OR) was derived by pooling each adjusted OR in the included studies. A subgroup analysis was performed based on the LCZ696 drug dose (100, 200, and 400 mg). Publication bias was determined by funnel plots; by visually inspecting the symmetry of the effect size distribution, and Egger’s test was also used for further confirmation. Fixed-effect or random-effect models were chosen according to heterogeneity, and inverse variance was used for weighted averages. We assessed the heterogeneity between studies using Cochran’s Q test and I^2^ statistics. Heterogeneity was interpreted as significant if I^2^ > 50%. All calculations were performed using R (ver 4.0.0) with the metafor package (Version 2.4-0) (Viechtbauer 2017), with a *p* < 0.05 taken as statistically significant.

## 3. Results

### 3.1. Enrollment of Studies

The flow diagram of the study selection is shown in Figure 1. In total, we identified 287 studies. Of these, 217 were deemed irrelevant after title and abstract screening, and 70 were assessed for eligibility using the full text. Of the 70 studies, 44 were excluded from the analyses after reading the title and abstracts. In the remaining 26 studies, 16 studies were excluded because of the study design, and the outcomes have not been published. Finally, 10 studies were included in the quantitative synthesis. The 10 studies were conducted in a number of different countries, including the U.S., Spain, Germany, U.K., Japan, Taiwan, and China [7,8,9,10,11,12,13,14,15,16]. A total of 5931 patients were randomized to receive either LCZ696 (at doses ranging from 100 to 400 mg per day) or a comparator drug (olmesartan) in five studies [8,11,13,14], valsartan in three studies [7,10,12], amlodipine in one study [9], and placebo in one study. The duration of the studies ranged from 4 to 52 weeks. The trial design, treatment strategies, and safety and efficacy outcomes of the ten included RCTs are summarized in Table 1. All included RCTs were judged to be at a low risk of bias (Appendix A). Two authors, HFH and LCC assessed the risk of bias of all included trials. The kappa value was 0.78, *p* = 0.01.

### 3.2. Efficacy of the Antihypertensive Effect of Sacubitril/Valsartan (LCZ696)

Both the msSBP and msDBP were calculated in all included studies. The systolic and diastolic blood pressure levels significantly decreased from the baseline after LCZ696 therapy (Figure 2). The pooled weight mean difference (WMD) demonstrated that the blood pressure reductions achieved with LCZ696 were more profound than those found after therapy with olmesartan, valsartan, or placebo. Compared with the comparator therapy (olmesartan in five studies, valsartan in three studies, or amlodipine in one study) or placebo, LCZ696 showed a significant reduction in msSBP with WMD at −6.52 mmHg (95% confidence interval (CI): −8.57 to −4.47; *p* < 0.001); while the WMD for msDBP was −3.32 mmHg (95% CI: −4.57 to −2.07; *p* < 0.001). Thereby, evidencing that LCZ696 has greater antihypertensive efficacy with respect to angiotensin receptor blockers or placebo in hypertensive patients at 24–52 weeks.

A total of ten studies explored the maSBP and maDBP from the baseline. These studies showed that LCZ696 is more efficacious than placebo in terms of reducing ambulatory systolic and diastolic blood pressure. Compared with the placebo therapy, LCZ696 showed a significant reduction in maSBP with WMD = −7.08 mmHg (95% CI: −10.48 to −3.68; *p* < 0.001), and maDBP with WMD = −3.57 mmHg, (95% CI: −5.71 to −1.44, *p* < 0.001) (Figure 3).

### 3.3. Effects of Different Doses of Sacubitril/Valsartan (LCZ696) Versus the Placebo Group

In the 100 mg dose of LCZ696 with comparators, the result did not show a difference in the msSBP reduction with WMD at −6.55 mmHg (95% CI: −16.89 to 3.79; *p* = 0.21). In the 200 mg dose of LCZ696 with comparators, the result showed a significant reduction in msSBP with WMD at −6.64 mmHg (95% CI: −9.62 to −3.66; *p* < 0.001). In the 400 mg dose of LCZ696 with comparators, the result showed a significant reduction in msSBP with WMD at −6.41 mmHg (95% CI: −9.53 to −3.28; *p* < 0.001) (Figure 2). In the 100 mg dose of LCZ696 with comparators, the result showed no difference in msDBP reduction with WMD at −4.29 mmHg (95% CI: −11.16 to 2.57; *p* = 0.21). In the 200 mg dose of LCZ696 with comparators, the result showed a significant reduction in msDBP with WMD at −3.47 mmHg (95% CI: −5.18 to −1.76; *p* < 0.001). In the 400 mg dose of LCZ696 with comparators, the result showed a significant reduction in msDBP with WMD at −2.90 mmHg (95% CI: −4.73 to −1.08; *p* < 0.01) (Figure 2).

Similar observations were also seen for maSBP and maDBP (Figure 3A,B).

### 3.4. Adverse Effects of Sacubitril/Valsartan (LCZ696)

Several drug-related AEs were reported after therapy in nine studies. The pooled data showed no significant difference between the AE rates of LCZ696 and the placebo (OR = 1.02, 95% CI: 0.83–1.27, *p* = 0.90) (Figure 4). In subgroups, 100 mg of LCZ696 showed an odds ratio of 0.94 (95% CI: 0.49–1.81, *p* = 0.86); 200 mg of LCZ696 showed an odds ratio of 1.12 (95% CI: 0.89–1.41, *p* = 0.34); while 400 mg of LCZ696 had an odds ratio of 1.06 (95% CI: 0.86–1.32, *p* = 0.57). There was also no significant difference between all subgroups according to different LCZ696 doses. The commonly reported AEs in the LCZ696 groups were nasopharyngitis, headache, dizziness, upper respiratory tract infection, diarrhea, and hyperuricemia. All of which were minimal and mild.

There were two cases of angioedema in the LCZ696 groups and one case in the olmesartan group, all of which were mild and resolved without hospitalization. Except for angioedema, there was one case of acute hepatitis that was attributed to LCZ696 [8], but the patient recovered spontaneously. Drug discontinuation due to medical intolerance was comparable, at about 1% in each group.

### 3.5. Adjunctive Evaluations Concerning the Risk of Publication Bias and the Stability of Results

All studies had randomized controlled and double-blind trials, and all the prespecified outcomes were reported. The ten trials were parallel group, multicenter trials. Overall, the included studies were of high quality. The Egger’s test revealed that there was a publication bias that existed in the estimates of pooled effect size of msSBP, but not in other outcomes. (Figure 5).

## 4. Discussion

In this meta-analysis, we found that LCZ696 reduced blood pressure more efficaciously than traditional therapy in msSBP, msDBP, maSBP, and msDBP, without increasing adverse events in hypertensive patients.

### 4.1. Antihypertensive Effectiveness of LCZ696

In our meta-analysis, we explored the vasodilation and anti-hypertensive properties of LCZ696, based on a comparison with two comparator drugs belonging to the ARB class (olmesartan and valsartan). We found that LCZ696 showed consistently better blood pressure control than traditional antihypertensive agents. There was a concordance in the blood pressure reduction in the LCZ696 group, including msSBP, msDBP, maSBP, and maDBP. Compared with the traditional angiotensin receptor inhibitor, which mainly acts to reduce the sympathetic drive, or inhibition of the renin-angiotensin-aldosterone system, LCZ696 inhibits both the renin-angiotensin-aldosterone system and the breakdown of serum natriuretic peptides, which results in further natriuresis and vasodilation [12].

Our meta-analysis demonstrated that LCZ696 has a stronger blood pressure-lowering efficacy in msSBP and msDBP, compared with the corresponding dose of valsartan, olmesartan, amlodipine, or placebo. The blood pressure-lowering capability was dose-dependent, and the blood pressure control rate also increased with higher doses of LCZ696, achieving 54.2% in patients receiving under 400 mg of LCZ696 [8]. Compared to the placebo and 320 mg valsartan, LCZ696 could decrease the systolic blood pressure by approximately 12 mmHg [8] and 8 mmHg [12], respectively. The blood pressure-lowering effect of LCZ696 started in the first week and remained stable throughout the observation period in all articles.

Ambulatory blood pressure measurement also showed a more efficacious blood pressure-lowering capability in the LCZ696 group, and the reduction was dose dependent. The general effects of blood pressure lowering with LCZ696 were maintained over the full 24-h dosing interval, with significantly greater reductions at night time [13]. The nighttime ambulatory systolic blood pressure reduction in Asian people also seemed to be greater than that in Westerners [8,10]. Kario et al. speculated that LCZ696 would contribute to reducing nighttime blood pressure through an increased NP-mediated mechanism in Asians. Because nighttime blood pressure levels are more closely related to the cardiovascular prognosis than daytime blood pressure, there may be more cardiovascular benefits for Asians using LCZ696 for treating hypertension. The ambulatory PP reduction was also more significant in the LCZ696 group [8,10].

### 4.2. Mechanisms of LCZ696

LCZ696 inhibits both natriuretic peptide degradation and angiotensin 1 receptor 1 (AT_1_ receptor). The natriuretic peptide system plays a key role in the pathophysiology of the cardiovascular system, especially the atrial natriuretic peptide and brain natriuretic peptide [21]. Neprilysin inhibitors accelerate vasodilation and natriuretic activity by preventing the breakdown of BNP and bradykinin. However, neprilysin inhibition also increased the levels of angiotensin I and II peptides [22]. This may explain the variable effects of neprilysin inhibitors on blood pressure, which can be unchanged, increased, or decreased in hypertensive people. It has been demonstrated that the renin-angiotensin-aldosterone system (RAAS) also has a “protective effect” by stimulating the angiotensin 2 (AT_2_) receptor [23]. The vasodilation effects of the AT_2_ receptor stimulation can be produced if the AT_1_ receptor is blocked [24]. Therefore, LCZ696 combines neprilysin inhibition with an angiotensin receptor blocker, increasing the levels of Ang II that tends to bind to AT_2_ receptors, and produces a positive biological response. These mechanisms appear to be responsible for the greater anti-hypertensive effect of LCZ696 compared to ARBs alone.

### 4.3. Adverse Effects of Sacubitril/Valsartan (LCZ696) in Hypertensive Patients

LCZ696 is generally safe and well tolerated. The adverse events between all groups were low and comparable. The most common adverse events in the LCZ696 group were diarrhea and nasopharyngitis [13]. Other recorded adverse events in the LCZ696 groups included appendicitis, urinary calculus, and cholelithiasis, which each contributed to only one case in all patients. The incidence rate of discontinuation due to adverse effects was low (~1%) and showed no difference between LCZ696 and other groups. There were four cases of angioedema [10,13,14], all of which were mild in nature and spontaneously recovered without hospitalization. Three occurred in the LCZ696 group and one occurred in the valsartan group. All deaths were not considered to be related to LCZ696 by the investigators.

Regarding the side effects, LCZ696 showed good tolerance in all subgroups. The most frequently reported adverse effects in the LCZ696 group were nasopharyngitis, upper respiratory tract infection, and dizziness, which were also equally reported in the placebo group. There was comparable safety and tolerability with an angiotensin receptor blocker or placebo. There were two cases of angioedema in the LCZ696 group and one in the olmesartan group, which were all mild and then recovered without hospitalization. All deaths (including one cardiovascular death) in the LCZ696 group were later confirmed to be unrelated to the LCZ696 treatment. All trials showed good tolerability, even when compared with placebo. There was also a comparable rate of drug discontinuation (approximately 1% per group) in both groups. Regarding serious adverse events, only one acute hepatitis case was attributed to LCZ696 because no other obvious etiology could be found. Generally, the adverse events of LCZ696 are few and it is well tolerated.

### 4.4. Clinical Implications

The significant blood pressure-lowering potency and acceptable adverse events of LCZ696 provide an effective and safe alternative for those whose blood pressure are still difficult to manage. Although there are many agents approved for lowering blood pressure, a potent single pill treatment may improve medical compliance. Improved blood pressure control also contributes to fewer cardiovascular events, given the enormous number of people with hypertension worldwide. All guidelines have set up blood pressure targets for different groups of people, and a more effective blood pressure-lowering capability will help many patients achieve their goal. Without a doubt, better blood pressure control will decrease the mortality and morbidity in cardiovascular disease.

### 4.5. Limitations

These trials also have several limitations. Firstly, their follow-up periods were generally short, from 28 days to 8 weeks. Endurance should be tested in further trials with longer follow-up periods. Second, these 10 studies only included a total of 5931 patients, which is relatively low compared to the massive number of patients that currently have hypertension. Third, all these studies included patients from different habitants, races, and ages, so these different inclusion criteria may be difficult to apply to all patients. Fourth, the traditional medication literature only included valsartan, olmesartan, and amlodipine, which could not represent all the currently approved medications. There are also no comparisons between LCZ696 with beta-blockers or diuretics. Fifth, the study groups only included essential hypertension, so the result could not be applied to the treatment of secondary hypertension. Sixth, there is still considerable heterogeneity in this study. We assessed the published year and LCZ696 dose as potential sources of heterogeneity and examined these factors as potential moderators in the meta-analysis; none was found to be significantly associated with the effect sizes. There may be other clinical or methodological differences among the studies that contributed to the statistical heterogeneity. However, we found that the main limitation of the study by Kario et al. was the lack of an active comparator [8]. Therefore, we tried excluding the Kario et al. study, and with reanalysis of the other nine RCTs in this meta-analysis, similar observations were seen for BP reduction with LCZ696, (Appendix A), but with improved heterogeneity of the study (Appendix A). Finally, this systematic review and meta-analysis study has not been registered, so it lacks transparency.

## 5. Conclusions

Sacubitril/valsartan (LCZ696) as a single daily dose may reduce blood pressure more efficaciously than traditional therapy (or placebo) in hypertensive patients without increasing adverse effects.

## Figures and Tables

**Figure 1 jcm-10-02824-f001:**
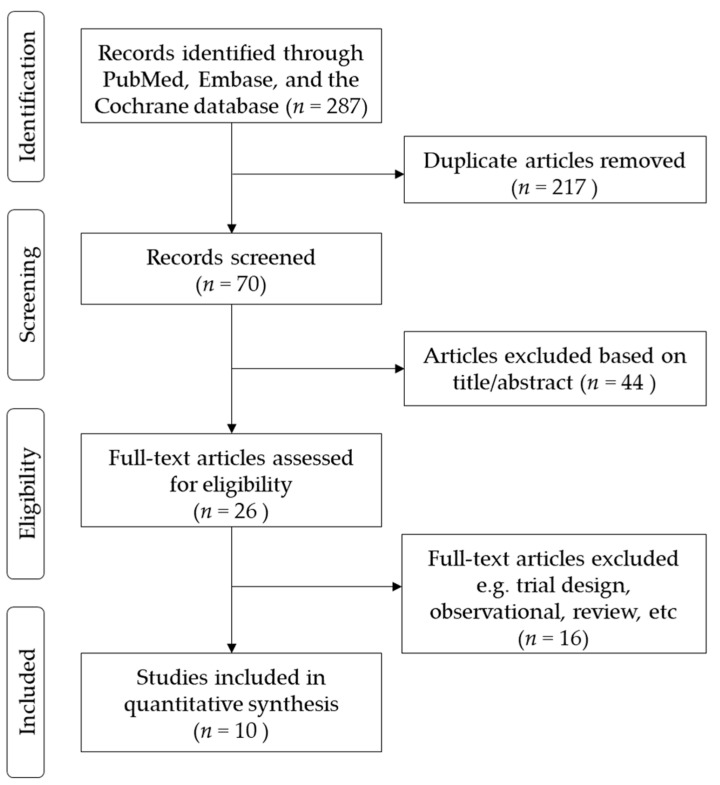
Flow diagram of included studies.

**Figure 2 jcm-10-02824-f002:**
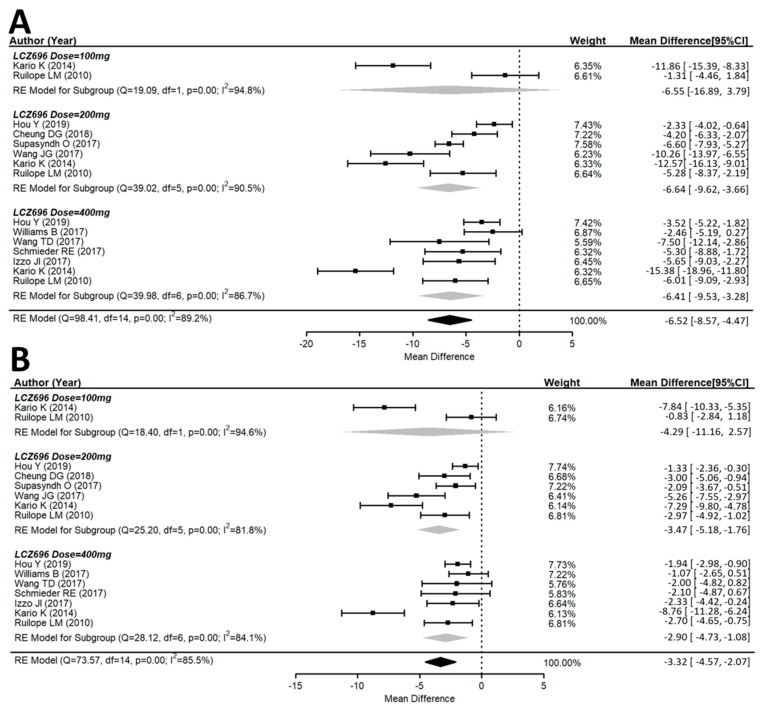
Forest plot of (**A**) msSBP and (**B**) msDBP. Comparisons of LCZ696 with a control group. msSBP, mean sitting systolic blood pressure; msDBP, mean sitting diastolic blood pressure.

**Figure 3 jcm-10-02824-f003:**
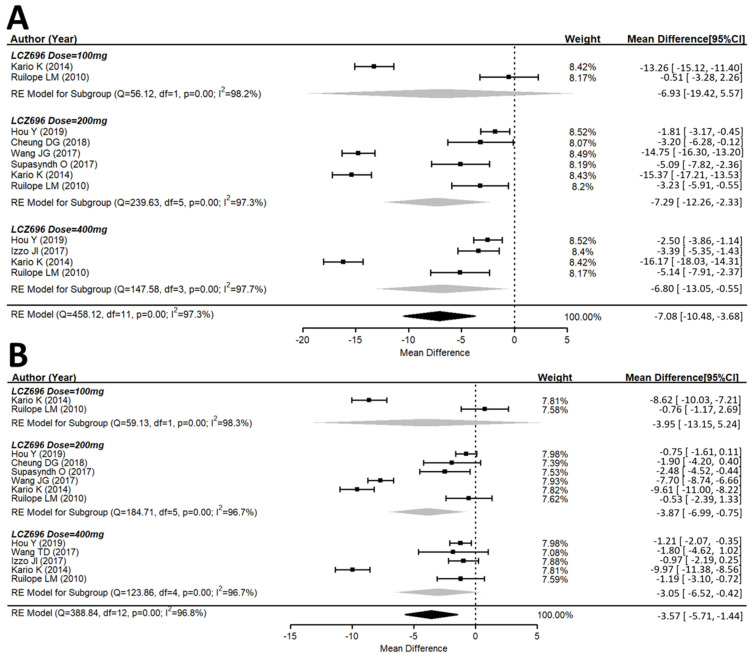
Forest plot of (**A**) maSBP and (**B**) maDBP. Comparisons of LCZ696 with a control group. maSBP, mean ambulatory systolic blood pressure; maDBP, mean ambulatory diastolic blood pressure.

**Figure 4 jcm-10-02824-f004:**
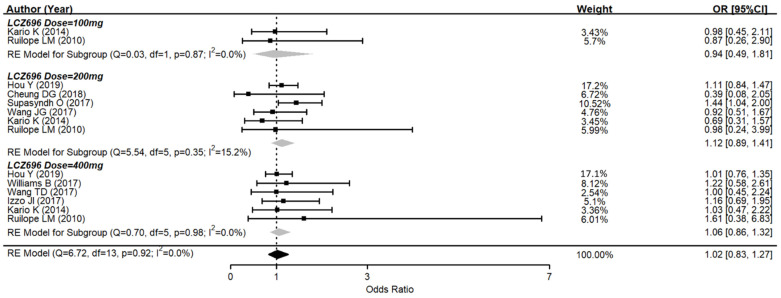
Forest plot of trial-defined adverse events. Comparisons between LCZ696 and a control group.

**Figure 5 jcm-10-02824-f005:**
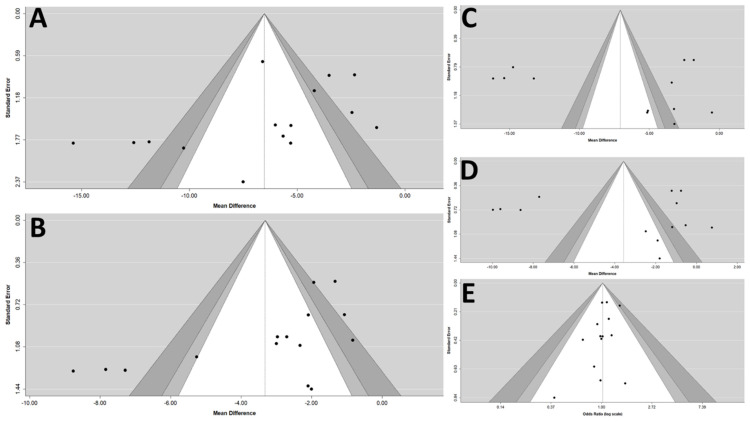
Funnel plots of (**A**) msSBP, (**B**) msDBP, (**C**) maSBP, (**D**) maDBP, and (**E**) adverse events. maSBP, mean ambulatory systolic blood pressure; maDBP, mean ambulatory diastolic blood pressure; msSBP, mean sitting systolic blood pressure; msDBP, mean sitting diastolic blood pressure.

**Table 1 jcm-10-02824-t001:** Characteristics of the included trials.

Author, Year	Race	Patients’ Characteristic	Age, Years	Number, (Female/Male)	LCZ696 Dose	Comparison Therapy	Observation Periods	Severe Adverse Events	Risk of Bias
Ruilope LM et al., 2010 [7]	not restricted	Mild-to-moderate essential hypertension	18–75	1328 patients (568/760)	100 mg, 200 mg, 400 mg	Valsartan 80 mg, 160 mg, 320 mg	8 weeks	Nil	3/3
Kario K et al., 2014 [8]	only Asians	mild-to-moderate essential hypertension	≥18	389 patients (114/275)	100 mg, 200 mg, 400 mg	Placebo	8 weeks	Nil	2/3
Wang JG et al., 2017 [9]	only Asians	mean clinic SBP 145–180mmHg	≥18	266 patients (110/156)	200 mg	Amlodipine 5 mg	8 weeks	Nil	3/2
Izzo JI et al., 2017 [10]	not restricted	mild-to-moderate systolic hypertension	≥18	907 patients (412/495)	400 mg	Valsartan 320 mg	8 weeks	1 angioedema and 1 CV death in LCZ696 group	3/3
Schmieder RE et al., 2017 [11]	not restricted	essential hypertension Stage 1 and 2	≥18	114 patients (37/77)	400 mg	Olmesartan 40 mg	52 weeks	NA	2/2
Wang TD et al., 2017 [12]	only Asians	patient with salt-sensitive hypertension	≥18	72 patients (26/46)	400 mg	Valsartan 320 mg	4 weeks	Nil	2/2
Williams B et al., 2017 [13]	not restricted	msSBP 140–179 mm Hg and PP > 60 mm Hg	≥60	454 patients (217/237)	400mg	Olmesartan 40 mg	12 weeks	NA	3/3
Supasyndh et al., 2017 [16]	only Asians	elderly Asian (≥65 years)	≥65	588 patients (294/294)	100 mg, 200 mg, 400 mg	Olmesartan 10 mg, 20 mg, 40 mg	14 weeks	Nil	3/3
Cheung et al., 2018 [15]	not restricted	mild-to-moderate hypertension	≥18	375 patients (183/192)	200 mg	Olmesartan 20 mg	8 weeks	Nil	2/3
Hou Y et al., 2019 [14]	only Asians	mild-to-moderate hypertension	≥18	1438 patients (682/756)	200 mg, 400 mg	Olmesartan 20 mg	8 weeks	1 Angioedema in each group	3/3

CV, cardiovascular; msSBP, mean systolic blood pressure in sitting position; PP, pulse pressure; Risk of bias, 3: low risk, 2: unclear risk, 1: high risk of bias.

## Data Availability

Data available on request due to restrictions.

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
