# Peer review of "The Antihypertensive Effects and Safety of LCZ696 in Patients with Hypertension: A Systemic Review and Meta-Analysis of Randomized Controlled Trials"

_jcm, 2021, doi:10.3390/jcm10132824_

Round 1

Reviewer 1 Report

Authors have performed a systematic metanalysis of  randomized clinical trials aimed to compare the efficacy and safety of sacubitril-valsartan versus other antihypertensive drugs or placebo.

A previous metanalysis , including only 5 studies and 1500 patients older than 65 years was performed in 2019.

The results show a slight reduction of BP (clinic and 24 hrs) during treatment with Arni, mainly when administered at dosages of 200 mg or higher as compared to other drugs.

Major comments

This is a nice review , including 10 studies

Methods are clearly described , and results are reported  in detail

The improvement in BP reduction observed with Arno as compared with a monotherapy is expected, since sacubitril -valsartan is a combination of 2 different drugs.

Only one study showed a significant reduction in both SBP and DBP during treatment with Arni at the dosage of 100 mg

(Kario). Authors should clearly explain that the reason was the comparison with placebo (Fig 1 and 2).

The comparison in the study by Izzo et al is shown only for the 400 mg dose. Perhaps it would be more informative keep comparisons vs placebo separated by comparisons vs all other drugs

Could authors try to analyse separately the antihypertensive effect by age groups (< or > 60 years), by sex (in men and women) and by ethnic origin ?

Author Response

We highly appreciate the reviewer 1’s insightful and helpful comments on our manuscript. Following are the responses to the reviewer 1’s comments point by point.

Only one study showed a significant reduction in both SBP and DBP during treatment with Arni at the dosage of 100 mg(Kario). Authors should clearly explain that the reason was the comparison with placebo (Fig 1 and 2).

Ans. We do thank the reviewer for the valuable comments. The main limitation of the Kario et al’s study was the lack of an active comparator. We try to exclude Kario et al study, and reanalysis the other 9 RCTs in this meta-analysis, similar observations were also seen for BP reduction with LCZ696, but improved heterogeneity of the study. We had stated this information in the supplementary Fig 2 to 5 and included these findings in the limitation of the revised article. (Line 284-293 of the revised article)

The comparison in the study by Izzo et al is shown only for the 400 mg dose. Perhaps it would be more informative keep comparisons vs placebo separated by comparisons vs all other drugs

Ans. Thank you for your suggestion. Of the 10 RCTs included in the present meta-analysis, 8 of the comparative drugs are angiotensin receptor blockers (5 olmesartan, 3 valsartan), amlodipine in a study (Wang JG et al), and placebo in 1 study (Kario et al). We have excluded Kario et al study and reanalyzed the other 9 RCTs in this meta-analysis. We observed similar observations of BP reduction effect with LCZ696. We had stated this information in the supplementary Fig 2 to 4 and limitation of the revised article. (Line 284 to 292 of the revised article) Thank you.

Could authors try to analyse separately the antihypertensive effect by age groups (< or > 60 years), by sex (in men and women) and by ethnic origin ?

Ans. Thank you for your suggestion. We do agree that if the antihypertensive effects of LCZ696 can be analyzed separately by age groups, gender, or race, it will be impressive. However, we do not have the raw data of each RCTs included in the present study. Thank you again.

Reviewer 2 Report

Systematic search processes are not clear/not standard and needs significant improvement: 

  1. This systematic reviews and meta-analysis has not been registered. Lack of transparency. Please add this point in the limitation.
  2. Search terms in Ovid Medline and Embase are different. Please attach search terms that were used in each database as supplement for Data source and search strategies in the manuscript. Please provide details search terms in supplementary documents.
  3. When Pubmed is used for the search, MESH terms are always recommended to be included.
  4. It will be better to show kappa for the selection and data extraction. Please show the data of kappa of agreement during the systematic searches. 
  5. Quality assessment needs to be provided with Cochrane Risk of Bias tool [for RCTs   
  6. Some revision of the English language is needed. There are some parts of the paper where it is quite difficult to make sense of some sentences English edit will help to improve the quality of the manuscript.
  7. Random or Fixed effect was used, needs to be specified in the abstract.
  8. Authors should discuss the reason of heterogeneity.
  9. There is still a considerable heterogeneity as in your limitation. Meta-regression analysis is then strongly recommended.
  10. Footnotes need to be provided below the Table

Author Response

Reviewer 2

We highly appreciate the reviewer 2’s insightful and helpful comments on our manuscript. Following are the responses to the reviewer 1’s comments point by point.

This systematic reviews and meta-analysis has not been registered. Lack of transparency. Please add this point in the limitation.

Ans. Thank you for your comment. Following your suggestion, we had included this information in the limitation of the revised article. (Line 293, revised article)

Search terms in Ovid Medline and Embase are different. Please attach search terms that were used in each database as supplement for Data source and search strategies in the manuscript. Please provide details search terms in supplementary documents.

When Pubmed is used for the search, MESH terms are always recommended to be included.

Ans. Thank you for your suggestion. The Study selection, search strategy and outcome measures were summarized in line 59 to 77 of the revised article. In addition, we also included the detail search terms in the supplementary documents. Thank you. (Line 59-77 of the revised article; supplementary Table in supplementary data)

It will be better to show kappa for the selection and data extraction. Please show the data of kappa of agreement during the systematic searches. 

Quality assessment needs to be provided with Cochrane Risk of Bias tool for RCTs   

Ans. Thank you for your suggestion. Two authors, HFH and LCC assessed the risk of bias of all included trials and prepared a risk of bias table as included in the supplementary Figure 1 of the revised article. Following your comment, we also included the value of kappa of agreement during the systematic searches in the method of the revised article. (Supplementary Figure 1; Line 110-111 of the revised article)

Some revision of the English language is needed. There are some parts of the paper where it is quite difficult to make sense of some sentences English edit will help to improve the quality of the manuscript.

Ans. Thank you for your comment. The revised article was edited by professional English editing service. Following please check the certificate of the English editing.

Random or Fixed effect was used, needs to be specified in the abstract.

Ans. Thank you for your correction. Following your suggestion, we had stated this information the revised abstract. “We performed a random-effects model meta-analysis of randomized controlled trial…”. (Line 14 of the revised article)

There is still a considerable heterogeneity as in your limitation. Meta-regression analysis is then strongly recommended.

Ans. Thank you for your recommendation. We assessed published year and LCZ696 dose as potential sources of heterogeneity and examined these factors as potential moderators in the meta-analysis, none was found to be significantly associated with the effect sizes. There may be other clinical or methodological differences among the studies contributed to statistical heterogeneity.

In addition, we found that the main limitation of the Kario et al’s study was the lack of an active comparator. Therefore, we try to exclude Kario et al study, and reanalysis the other 9 RCTs in this meta-analysis, similar observations were also seen for BP reduction with LCZ696, but improved heterogeneity of the study. We had stated this information in the supplementary Figure 2 to 5) and included these findings in the limitation of the revised article. (Supplementary Figure 2 to 5; Line 284-292, of revised article)

Footnotes need to be provided below the Table

Ans. Thank you for correction. We had included the footnotes below the Table in the revised article.

Round 2

Reviewer 1 Report

authors have addressed all comments raised by this reviewer 

Reviewer 2 Report

The investigators addressed my concerns the best the can. I have no additional concerns.